# Cold plasma-induced transcriptomic reprogramming and alternative splicing in tomato plants infected with ToBRFV

**Mahsa Rostami**[1]*, **Abozar Ghorbani**[1], **Davoud Koolivand**[2], **Abolfazl Mazandarani**[3]

1 Nuclear Agriculture Research School, Nuclear Science and Technology Research Institute (NSTRI), Karaj, Iran, 2 Department of Plant Protection, Faculty of Agriculture, University of Zanjan, Zanjan, Iran, 3 Plasma and Nuclear Fusion Research School, Nuclear Science and Technology Research Institute, Tehran, Iran

* msrostami@aeoi.org.ir

## Abstract

Cold atmospheric plasma (CAP), specifically cold air glow discharge plasma (CAGDP), offers a novel approach to enhancing plant defense against viral infections. This study investigates the effects of CAGDP on alternative splicing (AS) and transcriptome-wide gene expression in tomato seedlings infected with Tomato brown rugose fruit virus (ToBRFV). Using high-throughput RNA sequencing and bioinformatics analyses via CLC Genomics Workbench, custom Python scripts, and functional enrichment tools including STRING database and KEGG REST API, we identified significant AS changes, predominantly exon skipping and intron retention, in chromosomes associated with disease resistance. These splicing alterations were linked to key biological processes such as metabolic pathways, catalytic activity, and hormone signaling. Moreover, integration of miRNA–mRNA networks predicted by psRNA-Target and visualized in Cytoscape revealed a complex regulatory system involving both transcriptional and post-transcriptional mechanisms. The identification of 19 differentially expressed AS genes highlights the coordinated reprogramming induced by CAGDP to activate antiviral defenses. These findings highlight cold plasma technology as a promising, eco-friendly tool to reprogram plant molecular pathways and enhance resistance to viral pathogens.

## Introduction

Tomato (*Solanum lycopersicum* L.) is among the world's most widely cultivated vegetable crops, valued for its nutritional richness and economic importance. However, its productivity is increasingly threatened by viral pathogens, among which the Tomato brown rugose fruit virus (ToBRFV) has emerged as a devastating global concern. This virus is a rapidly spreading RNA virus primarily infecting tomatoes and peppers, causing significant crop losses worldwide. It spreads via contaminated seeds and

**Data availability statement:** All relevant data are within the paper and its Supporting Information files. The raw RNA-Seq data generated in this study have been deposited in the NCBI Sequence Read Archive (SRA) under BioProject accession number PRJNA1349013. The custom Python scripts used for alternative splicing event filtering are publicly available on GitHub at https://github.com/Abozarghorbani/Alternative-output-filtering.

**Funding:** The author(s) received no specific funding for this work.

**Competing interests:** The authors have declared that no competing interests exist.

mechanical contact during horticultural practices, with global seed distribution likely accelerating its spread. ToBRFV overcomes the resistance conferred by canonical R genes in tomato (Tm-1, Tm-2, Tm-2²) and pepper (L¹, L²), and no fully resistant tomato cultivars are currently available on the commercial market. Although recent studies have identified partial resistance sources—such as the Tm-1 allele from the wild relative Solanum habrochaites [1,2], quantitative trait loci (QTLs) in *S. pennellii* introgression lines [3], and CRISPR/Cas9-mediated knockout of eIF4E [4]—these approaches remain experimental and have not yet translated into field-ready, durable resistance. This persistent vulnerability underscores the urgent need for non-genetic, sustainable strategies to bolster plant defenses.

An effective method for controlling plant viruses involves using cold atmospheric plasma (CAP), with cold air glow discharge plasma (CAGDP) emerging as a promising option. This technique is recognized as an environmentally friendly and non-thermal technology that can deactivate various microbial contaminants, including viruses, bacteria, and fungi. Research shows that CAGDP not only neutralizes viral RNA but also promotes strong plant growth, providing dual benefits for increasing plant productivity and ensuring plant health [5]. In our previous study, a brief 3-minute exposure to CAGDP significantly decreased ToBRFV accumulation in tomato seedlings and also improved seedling vigor by enhancing germination, increasing chlorophyll content, and promoting greater stem diameter. These findings highlighted the promising antiviral and growth-promoting effects of plasma treatment. However, despite these positive phenotypic results, the molecular mechanisms underlying these responses, especially the transcriptome-wide changes and the role of post-transcriptional regulation, remain largely uncharacterized [6].

Alternative splicing (AS) in plants is a key mechanism that produces diverse transcripts from precursor mRNA, generating proteins essential for regulating various life activities. It influences processes such as seed germination, flowering, growth, development, and responses to environmental stresses. AS operates through a coordinated system of cis-regulatory elements and trans-acting factors on pre-mRNA, with splicing factors identifying cis-elements to manage splice site selection. Controlled by sequence variations, environmental signals, and different spliceosomes, AS ensures precise regulation, enabling plants to adapt dynamically to developmental and environmental changes [7].

In the realm of plant-virus interactions, AS plays a key role in regulating the expression of resistance genes, transcription factors, and signaling elements [8]. However, how CAGDP treatment influences AS patterns in ToBRFV-infected plants has not been studied. Although cold plasma has demonstrated efficacy in deactivating viruses and promoting plant vigor, its role in shaping the post-transcriptional landscape—particularly alternative splicing—remains unexplored in plant–virus systems. Notably, no transcriptomic study to date has examined how plasma-generated reactive oxygen species (ROS) influence splicing decisions in crops under viral stress. Emerging evidence suggests that ROS act as signaling molecules that can modulate the activity of splicing regulators through redox-sensitive cysteine residues, phosphorylation cascades, or chromatin remodeling, thereby altering splice site selection

and isoform diversity [9]. Given that AS is a rapid and reversible layer of gene regulation critical for immune responses, it is biologically plausible that CAGDP-induced ROS reprogram splicing networks to enhance antiviral defense— a hypothesis directly addressed in this study. Moreover, recent research has uncovered that plant viruses actively manipulate the host's splicing mechanisms to facilitate their infection processes. For instance, studies highlight that infection by the sugarcane mosaic virus alters alternative splicing patterns, ensuring sustained ZmPSY1 synthesis during infection, which enhances viral propagation [10]. Similarly, investigations into Potato Virus Y provide valuable insights into the interplay between virus infections and key processes like transcriptome reprogramming, RNA methylation, and alternative splicing, paving the way for future research [8]. On the other hand, host plants can leverage alternative splicing to create decoy receptor variants or truncated immune proteins that disrupt viral effectors [11]. These findings emphasize that alternative splicing serves as a critical battleground in the co-evolution of plants and viruses—an active regulatory arena shaped and targeted by both sides.

To elucidate the molecular basis of CAGDP's antiviral effects, we adopted a systems biology approach to investigate transcriptomic and post-transcriptional changes in ToBRFV-infected tomato seedlings. An extensive RNA-Seq analysis was performed to identify genome-wide AS events affected by plasma treatment. Using CLC Genomics Workbench along with custom Python pipelines, plasma-specific splicing modifications were identified and analyzed through functional enrichment, including Gene Ontology (GO) terms and KEGG pathways. To gain deeper insights into regulatory mechanisms, we explored potential interactions between alternatively spliced transcripts and tomato microRNAs (miRNAs). Additionally, differential gene expression (DGE) analysis was performed on splicing-related genes, revealing coordinated changes in transcription and splicing dynamics. This integrated approach provides insight into the molecular mechanisms behind plasma-driven antiviral responses, highlighting the importance of AS as a key regulatory component in tomato's defense against ToBRFV.

## Materials and methods

### Biological Material Processing and RNA Library Preparation

Tomato seeds of the cultivar SV 3725TH were extracted from fruits confirmed to be infected with ToBRFV by RT-PCR. The central flesh, where seeds are embedded, was carefully removed to extract seeds from infected fruits. The seeds were then separated, placed on filter paper under a laminar flow hood, and air-dried to reduce moisture. Dried seeds were stored in a dark environment at room temperature until treatment.

The CAGDP treatment was applied directly to dry seeds before sowing. The plasma system utilized a vacuum chamber equipped with two stainless-steel electrodes measuring 10 cm and 12 cm in diameter. The device operated under parameters of 6 kV, 50 W, and 25 kHz, employing ambient air as the working gas, with a sustained pressure level of $6.5 \times 10^{-2}$ Torr. Seeds were evenly distributed on a support tray positioned above the lower electrode and exposed to plasma for 3 minutes (designated as P3), based on our prior optimization [6]. Following treatment, seeds were sown and grown under controlled greenhouse conditions (day: 25–30°C; night: 16–20°C). At the four-leaf stage, total RNA was isolated from pools of 5 individual seedlings per biological replicate using TRIzol™ Reagent (Thermo Fisher Scientific, USA). Each experimental group, comprising P3 and the control, was represented by two biological replicates. The quality and integrity of the extracted RNA were verified through analysis with a NanoDrop spectrophotometer and agarose gel electrophoresis. Library construction was carried out using the Illumina TruSeq RNA Sample Preparation Kit, followed by sequencing on the Illumina NovaSeq 6000 platform, employing 150 bp paired-end reads. Subsequent data analysis for alternative splicing, including preprocessing, event classification, and functional annotation, was conducted as described in Fig 1.

### Bioinformatics workflow for alternative splicing and transcriptome analysis

**Read mapping and transcriptome assembly.** The raw sequencing reads were analyzed using version 25.0 of the CLC Genomics Workbench (QIAGEN). Following quality trimming to ensure a Q-score greater than 20, the reads were aligned to the tomato reference genome (SL4.0). BAM files resulting from this alignment were utilized for transcript reconstruction through the Transcript Discovery plugin, enabling the identification of both known and novel isoforms.

 

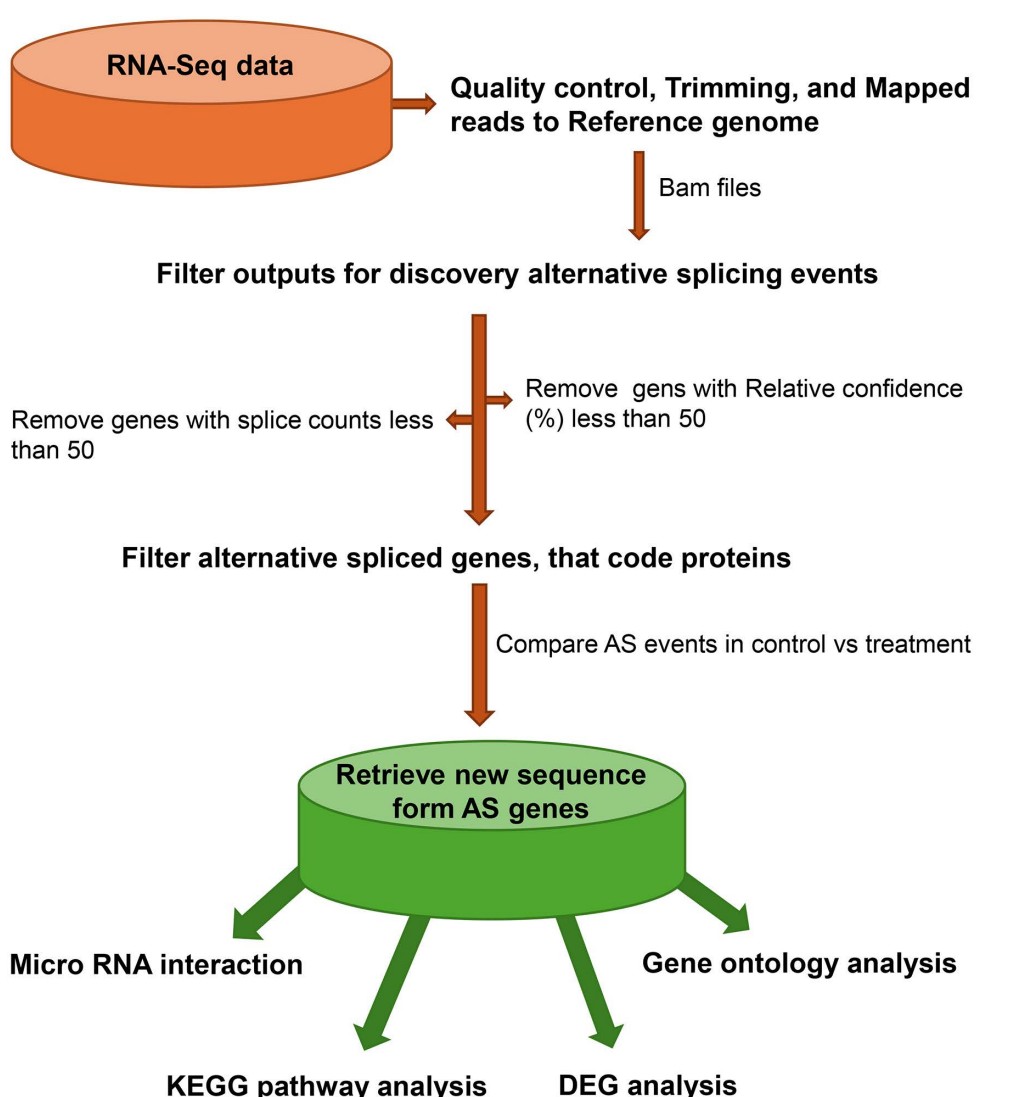

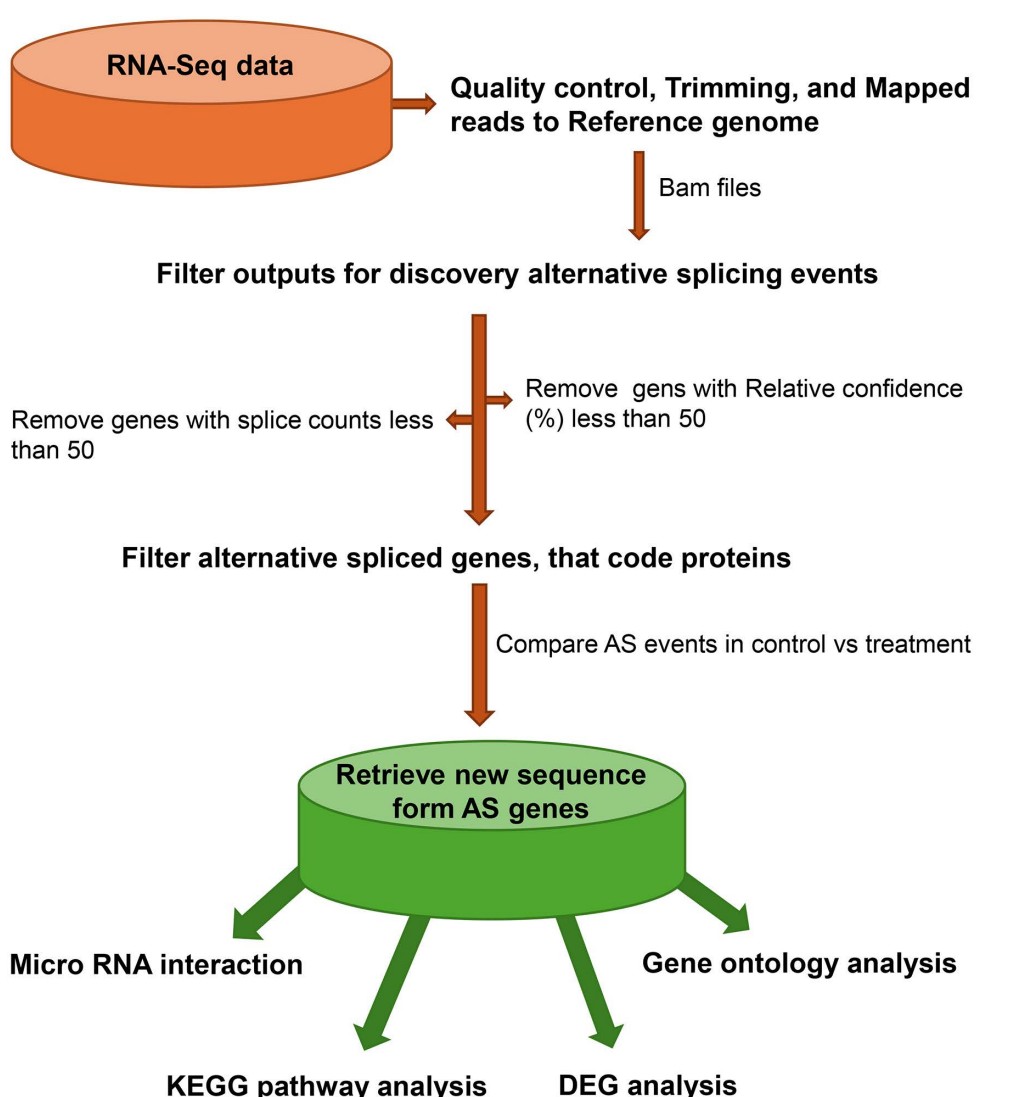

**Fig 1. Workflow of Alternative Splicing Analysis in Plasma-Treated and Control Tomato Plants Infected with ToBRFV.**

**AS event filtering criteria.** To ensure high-confidence detection of alternative splicing events, only splicing junctions supported by a minimum of 10 uniquely mapped reads and observed in at least one biological replicate (either plasma-treated or control) were considered for analysis. All genomic coordinates and gene annotations were based on the *S. lycopersicum* reference genome SL4.0 and the ITAG v4.1 gene annotation release.

**The AS event extraction and comparison.** To streamline the post-processing of AS data exported from CLC Genomics Workbench, a series of custom Python scripts was developed. Using the Pandas library, Excel output from both plasma-treated and control samples was parsed, cleaned, and converted into structured Data Frames containing key information such as gene identifiers, splice evidence, chromosomal coordinates, and splice site sequences. Excel files with multiple sheets were programmatically split and recombined to create a unified AS annotation table. A comparative pipeline was implemented to identify AS events occurring only in plasma-treated samples, only in controls, or in both conditions. To facilitate categorical analysis, a new column labeled Condition Presence was added to classify each event

as "plasma only", "control only", or "both". The final annotation tables were then exported as Excel files for downstream enrichment analysis, visualization, and interpretation (S1 Table).

Differential AS events between plasma-treated and control samples were identified using the Differential Splicing Analysis module in CLC Genomics Workbench.

This module calculates changes in exon and junction usage based on normalized read counts, applying the Empirical Analysis of DGE statistical model integrated in CLC.

Splicing events with FDR-adjusted p-value < 0.05 and a difference in exon/junction usage ≥10% were considered significant. This default configuration of the CLC pipeline was used to ensure analytical consistency

**Splicing type classification and visualization.** To classify and visualize AS events, a symbolic bar plot was generated using Python (https://github.com/Abozarghorbani/Alternative-output-filtering.git). Events were grouped into three main categories based on the Splicing description, New 5′ sequence, and New 3′ sequence fields: alternative 5′ splice site, alternative 3′ splice site, and other (representing complex or ambiguous splicing patterns). A custom symbolic mapping (e.g., "➤—⟶" to represent A5SS) was applied to annotate the resulting bar plots, which were constructed using the matplotlib library. Additionally, a Venn diagram was created with the matplotlib_venn package to illustrate the overlap of AS-associated genes between plasma-treated and control groups.

**AS-Region Sequence Extraction and FASTA Generation.** A custom Python workflow was developed to extract the genomic sequences corresponding to AS regions, leveraging the Biopython and pandas libraries. Genomic coordinates were parsed from region annotations (e.g., "complement (3212..3450)") and accurately mapped to the tomato reference genome using SeqIO. For genes located on the negative strand, reverse complement sequences were automatically generated. The extracted sequences were exported in both Excel and FASTA formats to facilitate validation, annotation, and downstream sequence-level analyses.

## Functional characterization of AS-Associated Genes

**Gene Ontology (GO) Enrichment Analysis.** Gene Ontology (GO) enrichment analysis was performed using the STRING database, focusing on three main categories: Biological Process (BP), Molecular Function (MF), and Cellular Component (CC). Enrichment significance was evaluated using the Benjamini–Hochberg false discovery rate (FDR) correction, with a significance threshold set at FDR < 0.05 to control for multiple testing.

**KEGG pathway analysis with python.** A standalone Python script was developed to perform automated KEGG pathway annotation for splicing-related genes. UniProt IDs were initially loaded from a text file and mapped to corresponding KEGG identifiers using the UniProt REST API. Pathway associations were then retrieved through the KEGG REST API (rest.kegg.jp). The workflow included built-in retry logic and error handling to ensure reliable and complete data acquisition. A frequency summary of associated pathways was generated and exported to Excel, while pathway names were dynamically retrieved and linked to their KEGG IDs. This automated approach enabled comprehensive functional annotation, providing biological context to genes affected by AS.

**The miRNA target prediction.** To investigate potential post-transcriptional regulation of AS genes, predicted interactions between *S. lycopersicum* miRNAs and AS gene transcripts were analyzed using the psRNATarget web server (https://www.zhaolab.org/psRNATarget), regarding miRBase v22. Only high-confidence interactions with expectation scores below 3 were considered for downstream analysis. Predicted miRNA–mRNA interaction networks were visualized in Cytoscape v3.9.1.

**The DEG analysis.** Differential expression analysis was performed using CLC Genomics Workbench. Genes were classified as differentially expressed alternatively spliced genes (DE-ASGs) if they satisfied two criteria: [1] a statistically significant expression change with an absolute fold change ≥ 2 and false discovery rate (FDR) < 0.05, and [2] the presence of AS events that were unique to either the plasma-treated or control condition (i.e., condition-specific splicing). This approach captures both plasma-induced and plasma-suppressed splicing events that are transcriptionally modulated.

## Results and discussion

### RNA-Seq profiling of tomato seedlings treated with CAGDP following ToBRFV infection

To explore transcriptomic alterations induced by cold plasma treatment in tomato plants infected with ToBRFV, RNA-Seq analysis was conducted on seedlings at the four-leaf stage obtained from infected seeds subjected to CAGDP for P3, alongside untreated infected controls. Virus detection via RT-PCR confirmed ToBRFV infection in all seed sources before plasma treatment. RNA was extracted from symptomatic tomato fruits with TRIzol™ (Thermo Fisher Scientific, USA), and cDNA was synthesized using a reverse transcription kit (Pishgam, Iran) with hexamer primers. PCR amplification utilized primers ToBRFV-F and ToBRFV-R [12] to produce an 842-bp RdRp fragment. Reactions used Taq DNA Polymerase Master Mix RED (Ampliqon, Denmark) on a Bio-Rad PTC-1148 cycler under specific thermal profiles. Amplicons were visualized on a 1% agarose gel, confirming all samples were ToBRFV-positive per a previous study [6].

Each sample produced approximately 35 million 150 bp paired-end reads. Quality assessment showed that over 95% of the reads passed filtering thresholds (Phred score >30), indicating high-quality sequencing data. The filtered reads were mapped to the *S. lycopersicum* SL4.0 reference genome, with alignment rates exceeding 93% across all samples. These metrics confirm excellent sequencing depth and mapping efficiency.

### CAGDP -Induced AS Events and Chromosome-Specific Responses to ToBRFV Infection in Tomato

Transcriptome analysis using the CLC Genomics Workbench and Transcript Discovery plugin revealed widespread AS events in both control and P3-treated samples. In total, 349 splicing events were identified. Notably, 74 splicing events were uniquely observed in the plasma-treated group, representing distinct transcript isoforms not detected in control seedlings. Only 16 splicing events are shared with the control (Fig 2a). Several genes exhibited more than one type of alternative splicing, indicating complex post-transcriptional regulation in response to CAGDP treatment. Additionally, among the five classical AS types, intron retention (IR) was the most prevalent, consistent with typical patterns in tomato. However, given the established roles of IR, exon skipping (ES), and mutually exclusive exons (MXE) in regulating immune-related gene function [9], we focused our biological interpretation on these three categories. Alternative 5′ and 3′ splice site events were also detected but were less frequent and not further analyzed due to limited evidence of their direct involvement in antiviral responses in tomato (Fig 2b). The presence of plasma-specific splicing suggests that cold plasma treatment influences splicing machinery and changes transcriptomic flexibility under viral stress. These findings support the idea that AS plays a key role in plant stress adaptation by producing functional transcript variants. These AS types have been previously linked to plant defense mechanisms, especially in genes involved in pattern-triggered immunity (PTI) and effector-triggered immunity (ETI). IR and ES are known to generate immune-related transcript variants in NLRs and PRRs, which are essential for pathogen recognition [9]. The fact that plasma specifically modifies these splicing patterns—particularly IR and ES, which are known to generate immune-related transcript variants in NLRs and PRRs — raises the hypothesis that CAGDP may enhance immune-related mRNA diversity, similar to pathogen-induced AS. While it is plausible that such splicing changes could modulate defense-related gene expression or contribute to a primed state, this interpretation remains speculative in the absence of functional validation of the resulting isoforms or direct evidence of accelerated immune responses. Future studies involving isoform-specific assays and secondary viral challenges will be needed to test whether plasma-induced AS functionally contributes to antiviral priming.

Fig 3 displays all ASs across all chromosomes in the P3 treatment. It indicates that ASs are found throughout the genome and are especially abundant on chromosomes 1, 3, and 9. In this study, a notable increase in AS events was observed on chromosome 9 of *S. lycopersicum*, which may have biological significance related to viral defense. Chromosome 9 is of particular interest because it harbors the well-characterized Tm-2 resistance gene (Solyc09g018220), which confers resistance to several tobamoviruses, though it is ineffective against ToBRFV [1,2]. Similarly, chromosome 3 contains a quantitative resistance locus previously associated with ToBRFV response in tomato breeding lines [3]. While our

## Alternative Splicing Gene Comparison (Plasma vs Control)

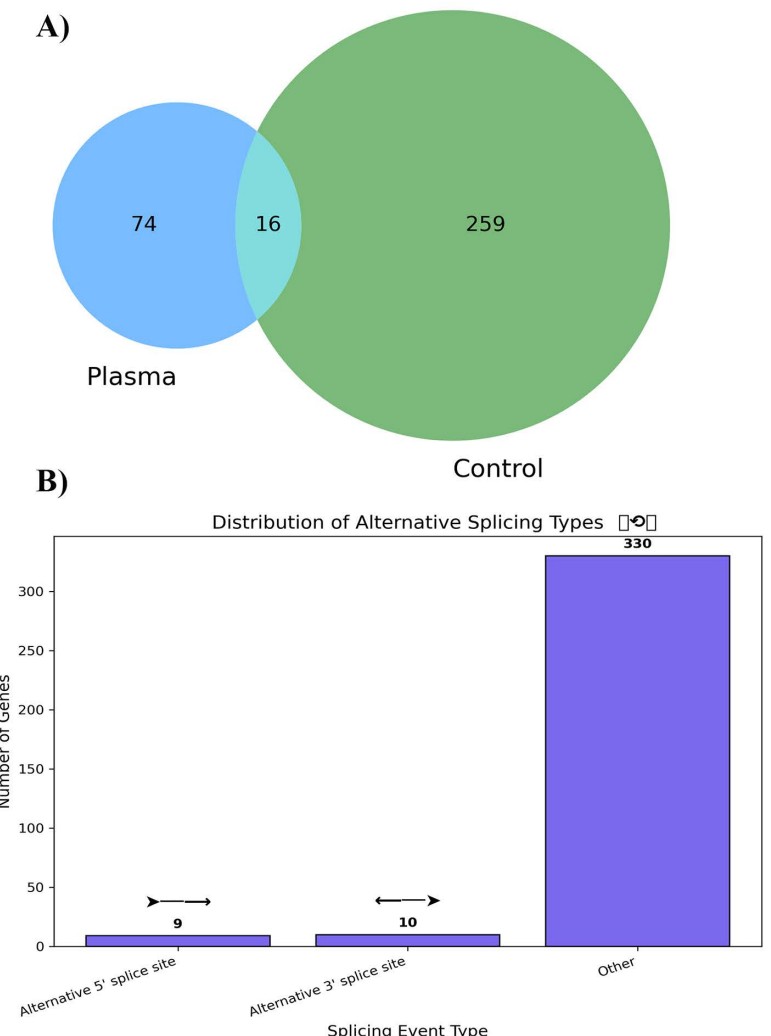

**Fig 2. Plasma-induced alternative splicing patterns in tomato seedlings infected by ToBRFV. (a)** Venn diagram comparing AS events between plasma-treated (P3) and control seedlings. **(b)** Distribution of AS event types among plasma-specific genes.

AS annotation did not identify splicing events directly within Tm-2 or other canonical R-genes, this also applies to Tm-1, for which no alternative splicing was detected under plasma treatment. The elevated density of plasma-induced AS events in these genomic neighborhoods suggests that regulatory or structural genes in proximity to resistance loci may undergo post-transcriptional modulation. Such modulation could indirectly influence defense signaling or chromatin accessibility in these regions. However, we emphasize that this remains a hypothesis—co-localization does not imply causation—and functional validation will be required to establish any mechanistic link. Furthermore, chromosome 1 is crucial in tomato, housing important regulatory genes like SlMYB12, which controls fruit peel coloration through flavonoid biosynthesis [1,2]. Recent transcriptomic studies have shown that chromosome 1 exhibits one of the highest frequencies of AS events in tomato [13]. In our research, the prominent occurrence of plasma-induced AS events on this chromosome may reflect its role in regulating stress responses under ToBRFV infection.

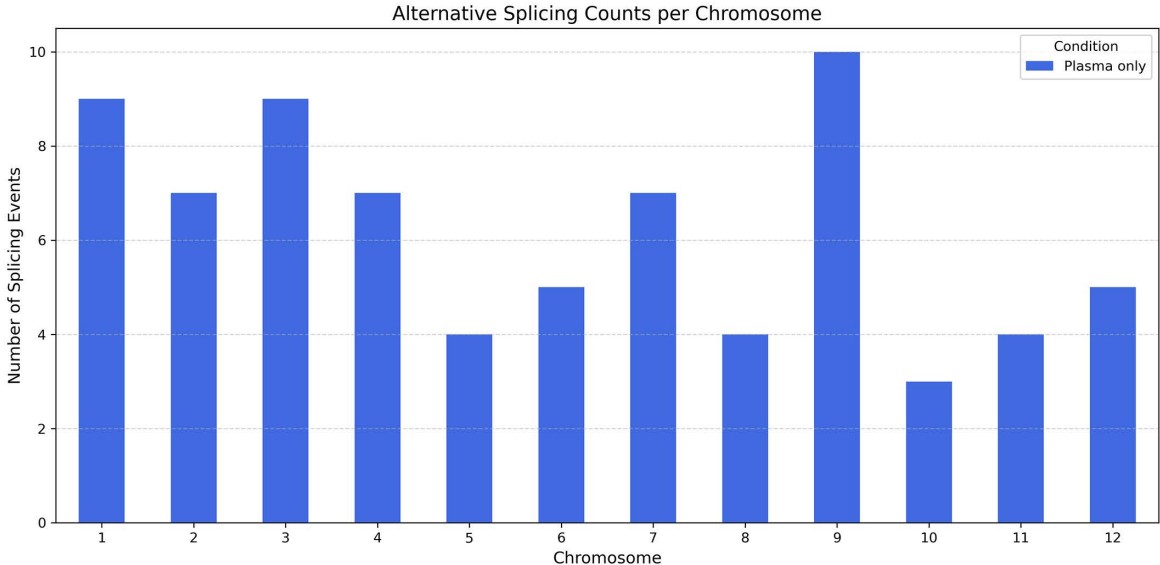

Fig 3. Chromosomal distribution of plasma-specific alternative splicing events in tomato seedlings infected by ToBRFV.

## Functional enrichment of plasma-induced alternatively spliced genes

To elucidate the functional implications of plasma-induced AS events, a GO enrichment analysis was conducted, encompassing three primary categories: MF, BP, and CC (Fig 4).

In the MF category, two terms were significantly enriched: catalytic activity acting on proteins and transferase activity (Fig 4a). These functions are closely linked to post-translational modifications and cellular signaling processes, reflecting the sustained activation of effector enzymes that are crucial for modulating plant responses to environmental stimuli. For instance, an upregulation of catalytic activity has been documented in tomato plants subjected to salt stress, where it enhances metabolic flexibility and stress tolerance [14]. Moreover, transferase enzymes, such as Glutathione S-Transferase, play a vital role in detoxification and stress responses by neutralizing harmful compounds, regulating plant growth, and mediating adaptive responses to various environmental stresses [15]. The observed enrichment of these activities suggests that AS may influence enzymatic or regulatory proteins under cold plasma treatment, potentially fine-tuning stress and immune-related pathways through modulation of protein functions.

In the BP category, AS-associated genes were highly enriched in terms related to metabolism, including metabolic process, primary metabolic process, organic substance metabolic process, and nitrogen compound metabolic process (Fig 4b). The enrichment of genes associated with AS in metabolic processes underscores its pivotal role in regulating plant metabolism during stress conditions. By affecting essential pathways like metabolism, AS plays a vital part in the dynamic restructuring of biochemical networks required for adapting to stress. This ability to adjust metabolic processes not only helps maintain energy balance but also empowers plants to refine their physiological responses, enhancing their capacity to withstand environmental challenges [16]. The significant presence of these terms indicates that cold plasma treatment may influence the plant's metabolic reprogramming under viral stress through AS. Besides, the spatial distribution of AS in cells (Fig 4c) suggests that AS events influence gene products located in multiple key subcellular compartments, especially organelles involved in biosynthesis and transport. Research has shown that the subcellular localization of auxin biosynthesis enzymes, including YUCCA family members, is precisely regulated and crucial for proper hormone distribution and function. These findings emphasize the importance of AS in directing protein targeting within different cellular compartments, thereby impacting biosynthetic and transport pathways [17].

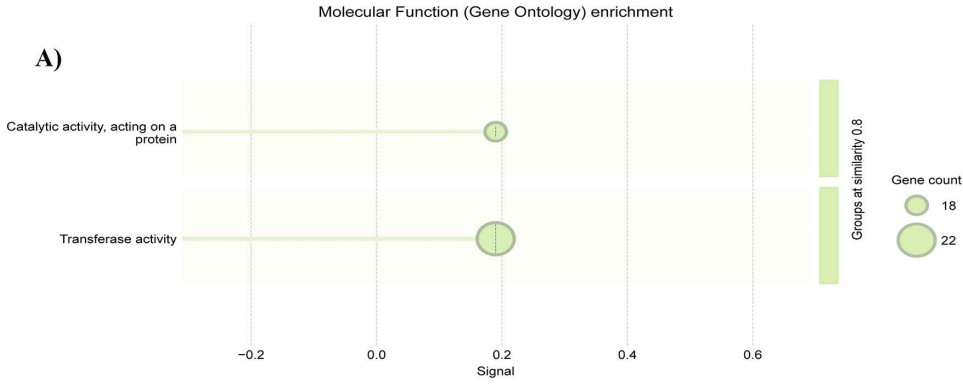

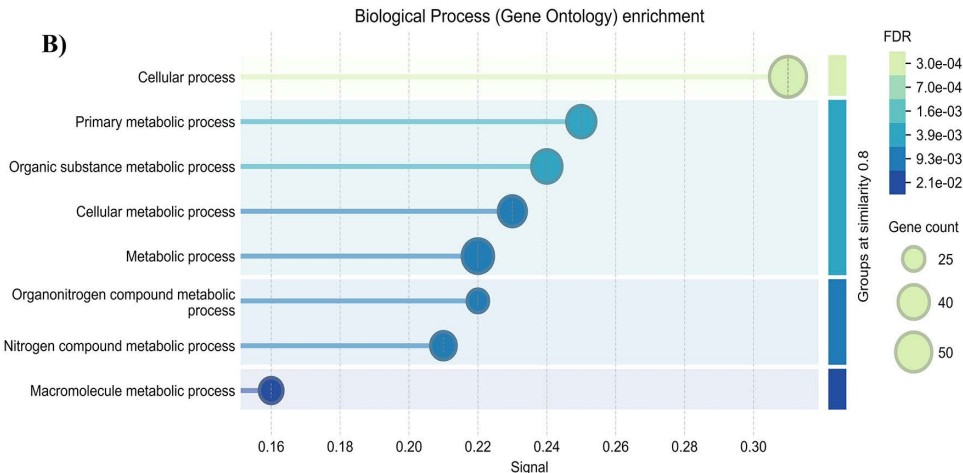

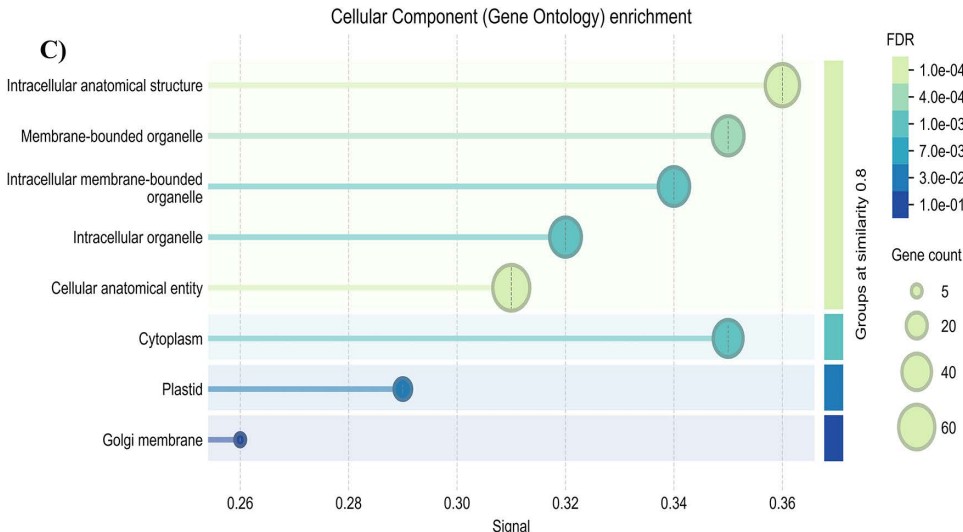

**Fig 4. Gene Ontology (GO) enrichment analysis of genes exhibiting plasma-specific alternative splicing events in plant tomatoes infected by ToBRFV (FDR<0.05).**

## KEGG pathway enrichment

The KEGG analysis showed that genes with AS events during plasma treatment are linked to 17 different pathways (Fig 5). Notably, the most enriched pathways were metabolic pathways, biosynthesis of secondary metabolites, and carbon metabolism, each with the highest gene counts. The notable increase in metabolic and secondary metabolite biosynthesis pathways among AS-associated genes indicates that plasma exposure may induce reprogramming of plant metabolic networks at the post-transcriptional level. Since AS directly influences secondary metabolism by regulating primary metabolite flow and hormonal balance [18], this response likely helps the plant better adjust its metabolite profiles to survive under plasma-induced stress conditions. Additional pathways, including RNA polymerase, spliceosome, and plant hormone signal transduction, underscore the regulatory importance of AS in transcriptional processes and hormone-driven stress response mechanisms. Interestingly, the KEGG analysis revealed enrichment of plasma-induced AS events in the ubiquitin-mediated proteolysis pathway. Several genes encoding E3 ubiquitin ligases and proteasome subunits exhibited condition-specific splicing patterns exclusively in plasma-treated seedlings. Since the ubiquitin-proteasome system governs the selective degradation of regulatory and damaged proteins during stress, AS-mediated modulation of its components could influence the stability of key immune signaling proteins. For example, a truncated E3 ligase isoform generated by intron retention might act as a dominant-negative regulator, thereby prolonging the activity of defense-promoting transcription factors [19]. While functional validation is needed, these findings suggest that CAGDP may reshape protein homeostasis networks not only transcriptionally but also through splicing-dependent mechanisms.

## miRNA–mRNA Regulatory Interactions

The miRNAs play a key role in the post-transcriptional regulation of gene expression. These endogenous, small non-coding RNA molecules are typically 20–24 nucleotides in length and play a crucial role in various regulatory processes within cells. A single miRNA can target multiple genes, and conversely, several different miRNAs may regulate the same gene. Their regulatory functions include inhibiting mRNA translation, promoting mRNA degradation, or facilitating poly A

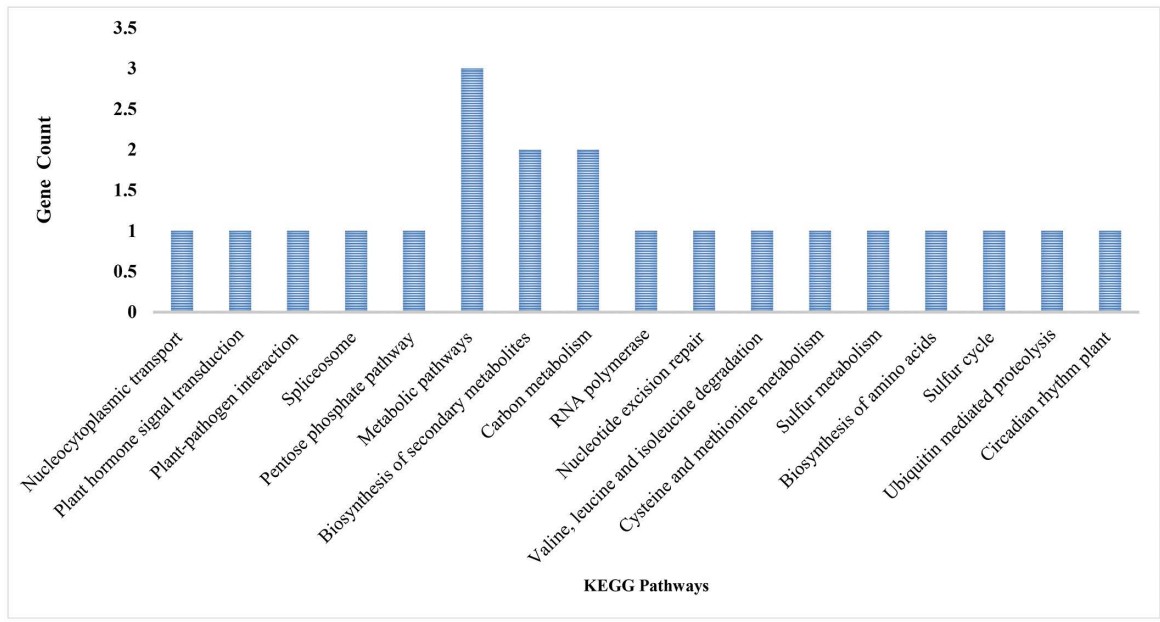

**Fig 5. KEGG pathway enrichment of plasma-specific alternatively spliced genes in the tomato plant.** –log10 adjusted p-value.

tail shortening [20]. Evidence indicates that tomato miRNAs are involved in the development of shoots, flowers, and fruits, as well as in regulating responses to both biotic and abiotic stresses [21]. One of the main objectives of this study was to identify miRNAs that may regulate the introduced AS gene, particularly under plasma treatment conditions. To explore potential post-transcriptional regulatory mechanisms, predicted interactions between tomato miRNAs and the AS gene were analyzed using the psRNATarget platform. These interactions, specific to the plasma-treated samples, were visualized using Cytoscape (Fig 6) (S2 Table).

Among the AS genes, Solyc12g056650.2, Solyc09g098240.3, and Solyc01g060470.3 are targeted by more miRNAs. Also, the most frequently replicated miRNAs in our analysis—sly-miR395a, sly-miR395b, and sly-miR319a—play crucial roles in regulating metabolism, developmental processes, and stress responses in tomato. The miR395a/b targets genes involved in sulfate metabolism and ATP sulfurylase activity, thereby modulating redox homeostasis and enhancing pathogen resistance—as demonstrated in rice against Xanthomonas oryzae [22]. In tomato, this conserved miRNA likely fine-tunes sulfur assimilation pathways under viral stress, potentially influencing defense-related metabolite production. Furthermore, the miR319 family, a phylogenetically ancient and highly conserved regulator in plants, orchestrates developmental programs and stress adaptation. To dissect the mechanistic basis of sly-miR319c during *Botrytis cinerea* infection, engineered transgenic Arabidopsis lines overexpressing either sly-miR319c or its cognate targets were used. Strikingly, miR319c-overexpressing plants exhibited significantly enhanced resistance to *B. cinerea*, implicating this miRNA as a key modulator of fungal defense pathways [23]. In tomato, miR319a plays a central role in orchestrating the plant's defense mechanisms against biotic stressors. By targeting TCP transcription factors, miR319a regulates jasmonic acid (JA)-dependent signaling pathways, particularly during the dual interaction with the beneficial fungus *Funneliformis mosseae* and the pathogenic cucumber mosaic virus (CMV). The interaction between this miRNA and associated long non-coding RNAs (lncRNAs) carefully balances immune responses, boosting resistance to viral infections while preserving beneficial symbiosis [24]. These results collectively suggest that CAGDP-induced transcriptome reprogramming in tomato under viral stress is multi-layered, involving splicing, expression modulation, and miRNA regulation.

## Integration of AS and DEG Analyses

To deepen the understanding of transcriptomic changes in response to CAGDP, we integrated differential expression with condition-specific AS events. A total of 19 DE-ASGs were identified: 8 exhibited plasma-specific AS (P-DE-ASGs), while 11 showed AS events exclusively in the control group (C-DE-ASGs), indicating that CAGDP may both induce novel splicing and suppress pre-existing splicing variants under viral infection (Table 1).

Among these, several genes were upregulated in plasma-treated samples, indicating activation in response to plasma exposure. For example, Solyc07g043390.3 (glycosyltransferase 2 family) exhibited a 6.46-fold increase in expression, possibly related to cell wall remodeling or metabolic adjustments. Glycosyltransferases are integral to various physiological processes in plants, including growth, development, and adaptation to biotic and abiotic stresses. Members of this enzyme family have been linked to enhanced resistance against *Verticillium* wilt in cotton [25]. In tomato, glycosyltransferases contribute to defense responses against *B. cinerea* and Tomato spotted wilt virus [26]. Additionally, their role is essential for conferring resistance to root-knot nematodes, highlighting their broad-spectrum function in plant immunity [27]. Conversely, Solyc03g005830.3 showed upregulation in control, possibly indicating suppression under plasma exposure.

Downregulated genes in plasma-treated seedlings included regulators such as Solyc07g006620.3 (protein kinase), Solyc02g083280.3 (rhodanese domain), and Solyc10g078285.1 (bZIP transcription factor), suggesting a reduction in specific signaling pathways that may serve to redistribute cellular energy for stress adaptation and immune response. Energy conservation is an important strategy that plants use to survive stressful conditions. Under environmental stress, plants often downregulate non-essential processes to conserve energy and divert resources for defence and survival. Research

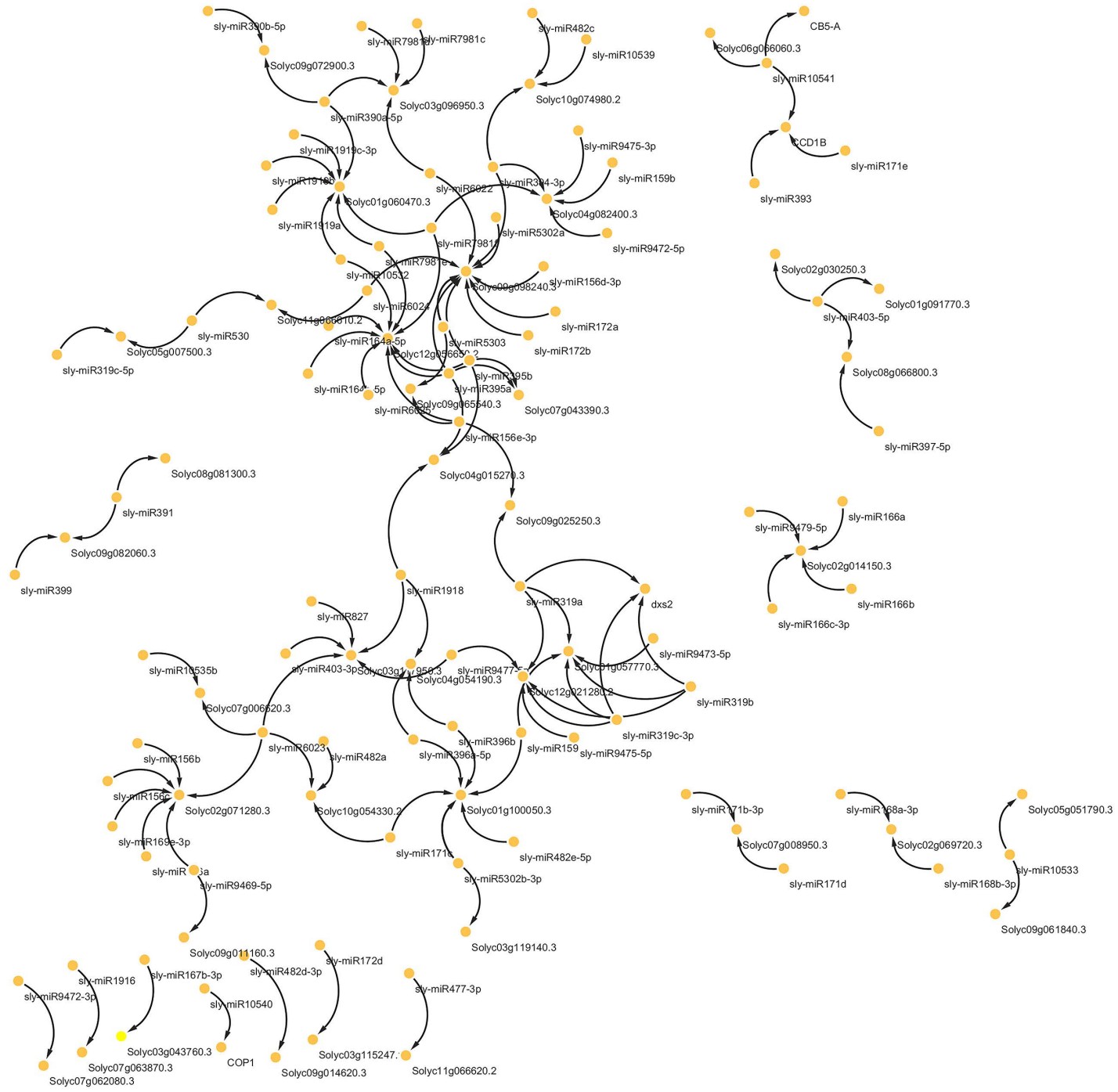

**Fig 6. Regulatory network of plasma-responsive tomato microRNAs and their alternatively spliced target genes under ToBRFV infection using psRNATarget.**

**Table 1. Differentially expressed and alternatively spliced genes identified in plasma (P3) and Control Samples under ToBRFV Infection.**

| Gene | Plasm/Control | Fold change | FDR p-value | Description |
|---|---|---|---|---|
| Solyc07g043390.3 | P | 6.46 | 0.03351663 | Uncharacterized protein; Belongs to the glycosyltransferase 2 family. (482 aa) |
| Solyc03g005830.3 | C | 2.59 | 0 | Cas1_AcylT domain-containing protein. (486 aa) |
| Solyc09g011160.3 | P | 2.46 | 0 | Uncharacterized protein. (494 aa) |
| Solyc12g009650.2 | P | 2.42 | 1.18823E-12 | AAI domain-containing protein. (265 aa) |
| Solyc09g065520.3 | C | −2.1 | 0.021604252 | Hcy-binding domain-containing protein. (343 aa) |
| Solyc05g051700.3 | P | −2.26 | 3.2962E-168 | Fe-S biosyn domain-containing protein. (176 aa) |
| Solyc06g066050.3 | C | −2.38 | 1.58758E-12 | Uncharacterized protein. (173 aa) |
| Solyc06g036305.1 | C | −2.43 | 6.19593E-65 | HTH myb-type domain-containing protein. (301 aa) |
| Solyc11g017460.2 | C | −2.48 | 2.44622E-10 | C2 domain-containing protein. (2157 aa) |
| Solyc02g083280.3 | P | −2.59 | 0.018046778 | Rhodanese domain-containing protein. (186 aa) |
| Solyc10g078285.1 | C | −2.69 | 0.00438144 | BZIP domain-containing protein. (822 aa) |
| Solyc07g006620.3 | P | −2.81 | 1.2288E-244 | Protein kinase domain-containing protein. (571 aa) |
| Solyc08g063050.3 | C | −2.83 | 1.68939E-21 | AAA domain-containing protein. (716 aa) |
| Solyc06g065800.3 | C | −3.78 | 0.04277328 | Uncharacterized protein. (236 aa) |
| Solyc10g045310.2 | C | −3.95 | 5.96467E-08 | Uncharacterized protein. (269 aa) |
| Solyc07g065480.3 | C | −5.57 | 0.000243118 | Uncharacterized protein. (142 aa) |
| Solyc09g047870.3 | C | −6.13 | 6.76707E-06 | FtsJ domain-containing protein. (204 aa) |
| Solyc06g068040.3 | C | −6.43 | 1.72311E-06 | Uncharacterized protein. (437 aa) |
| Solyc02g085140.3 | C | −21.78 | 2.0046E-05 | Unknown |

'P' denotes plasma-specific AS events; 'C' denotes control-specific AS events. All genes are differentially expressed ($|FC| \geq 2$, $FDR < 0.05$).

shows that under salt stress, plants adapt processes associated with photosynthesis- and energy metabolism- to maintain energy balance and improve stress tolerance [28].

The dual regulation at transcriptional and splicing levels implies a sophisticated reprogramming strategy involving both gene expression modulation and transcript isoform diversity. This subset of differentially expressed AS genes may thus act as central nodes in the CAGDP-triggered antiviral response, balancing activation and repression of signaling, metabolic, and regulatory processes.

## Limitations and Future Directions

While this study provides novel insights into the transcriptomic and splicing responses of tomato to cold plasma under viral stress, several limitations warrant further investigation. First, the analysis was restricted to a single susceptible cultivar (SV3725TH) and a single post-treatment time point (four-leaf stage). Although this stage was selected based on prior evidence of maximal antiviral effect and physiological response to CAGDP [6], it may not capture the full dynamics of splicing regulation over time. Future work should include longitudinal sampling and a panel of cultivars with varying genetic backgrounds—including those carrying known resistance genes (e.g., Tm-1, Tm-2²)—to determine whether the observed AS patterns are conserved or genotype-specific. Another major limitation of the present study is the lack of experimental validation of the functional consequences of the identified alternative splicing events. While our bioinformatic pipeline robustly detected plasma-specific isoforms and integrated them with expression and regulatory data, it remains unknown whether these splice variants produce stable proteins or alter protein function, localization, or interaction networks. Future studies should employ isoform-specific RT-PCR, proteomics, and targeted genetic approaches (e.g., CRISPR/Cas9 or VIGS) to validate the biological roles of key AS genes in mediating cold plasma-enhanced resistance to ToBRFV. Such functional validation will be crucial to translate these molecular insights into practical crop protection strategies.

While the miRNA–mRNA interaction network was constructed using stringent bioinformatic criteria, these predictions remain hypothetical without experimental confirmation. Future studies should employ degradome sequencing (e.g., PARE) to validate cleavage events, and transient expression assays to test the repressive effects of key miRNAs on their predicted alternatively spliced targets. Such validation will be essential to establish causal links between plasma-induced miRNA expression, AS regulation, and antiviral phenotypes. Furthermore, our findings provide compelling evidence that CAGDP can reprogram transcriptomic and post-transcriptional networks in tomato under viral stress, offering a sustainable strategy to enhance crop resilience. While this RNA-level analysis establishes a foundational molecular framework, future integration of proteomics, metabolomics, and functional validation will be essential to fully elucidate the causal links between plasma-induced splicing changes and antiviral phenotypes.

## Conclusion

This study highlights the significant impact of CAP, particularly CAGDP, on AS and gene expression in tomato seedlings infected with ToBRFV. The plasma treatment triggered distinct AS events—primarily exon skipping and intron retention—on key resistance-related chromosomes (1, 3, and 9), which are associated with metabolic, catalytic, and hormonal pathways. Moreover, integrated transcriptomic and miRNA analyses unveiled a complex regulatory network, identifying 19 differentially expressed and DE-ASGs that play pivotal roles in stress adaptation. Our findings provide compelling evidence that CAGDP can reprogram transcriptomic and post-transcriptional networks in tomato under viral stress, offering a sustainable strategy to enhance crop resilience and guide future antiviral breeding programs.

## Supporting information

**S1 Table. Genes exhibiting alternative splicing events in tomato plants treated with cold plasma and infected with ToBRFV.**
(XLSX)

**S2 Table. Predicted interactions between tomato miRNAs and the alternative splicing gene under cold plasma treatment.**
(PDF)

## Author contributions

**Formal analysis:** Abozar Ghorbani.

**Investigation:** Mahsa Rostami, Abozar Ghorbani, Abolfazl Mazandarani.

**Methodology:** Abozar Ghorbani, Abolfazl Mazandarani.

**Software:** Abozar Ghorbani.

**Supervision:** Abozar Ghorbani.

**Validation:** Abolfazl Mazandarani.

**Visualization:** Mahsa Rostami, Abozar Ghorbani.

**Writing – original draft:** Mahsa Rostami.

**Writing – review & editing:** Mahsa Rostami, Abozar Ghorbani, Davoud Koolivand, Abolfazl Mazandarani.

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
