## [Decision Letter · Decision Letter 0]

16 Oct 2025

Dear Dr. rostami,

Thank you for submitting your manuscript to PLOS ONE. After careful consideration, we feel that it has merit but does not fully meet PLOS ONE’s publication criteria as it currently stands. Therefore, we invite you to submit a revised version of the manuscript that addresses the points raised during the review process.

**ACADEMIC EDITOR:**

We look forward to receiving your revised manuscript.

Kind regards,

Shunmugiah Veluchamy Ramesh, PhD

Academic Editor

PLOS ONE

Journal Requirements:

3. In the online submission form, you indicated that [Insert text from online submission form here].

Additional Editor Comments:

Peer reviewers have provided feedback on your manuscript. While they recognize its potential, one reviewer has requested major revisions, and I concur with their assessment. Therefore, we invite you to revise the manuscript accordingly if you wish for us to reconsider it for publication.

Reviewers' comments:

Reviewer's Responses to Questions

**Comments to the Author**

1. Is the manuscript technically sound, and do the data support the conclusions?

Reviewer #1: Yes

Reviewer #2: Partly

2. Has the statistical analysis been performed appropriately and rigorously?

Reviewer #1: Yes

Reviewer #2: Yes

3. Have the authors made all data underlying the findings in their manuscript fully available?

Reviewer #1: Yes

Reviewer #2: Yes

4. Is the manuscript presented in an intelligible fashion and written in standard English?

Reviewer #1: Yes

Reviewer #2: Yes

Reviewer #1: Dear Author,

This study, titled "Cold Plasma-Induced Transcriptomic Reprogramming and Alternative Splicing in Tomato Plants Infected with ToBRFV," investigates the use of cold atmospheric plasma (CAP), specifically cold air glow discharge plasma (CAGDP), as an innovative approach to maintain plant defenses against the Tomato Brown Rugose Fruit Virus (ToBRFV) in tomato seedlings. The manuscript is clearly written and well-organized. However, it is primarily an in silico study, utilizing many bioinformatic tools. Although it offers novel insights into the effects of cold atmospheric plasma (CAP) on alternative splicing (AS):

- The study was performed on only one tomato cultivar (SV3725TH) and at a single time point (four-leaf stage seedlings) after treatment.

- The functional roles of the identified alternatively spliced genes and their corresponding isoforms still need to be validated through wet lab experiments (main drawbacks).

- Although miRNA-mRNA interactions were predicted bioinformatically using psRNATarget, an in vivo validation remains necessary.

- The study indicates that combining proteomics and metabolomics in future research could reveal the downstream effects of plasma-induced transcriptomic reprogramming and offer a more comprehensive understanding of plant-virus-plasma interactions. This shows that the current conclusion is incomplete. Minor typing errors, such as the "reference" (line 400), should be corrected. Additionally, the quality of the figures needs enhancement.

Reviewer #2: Praised the innovative application of CAP to address a critical agricultural problem and its focus on a nuanced molecular mechanism, as highlighted in the abstract

1. The abstract highlights significant AS changes, predominantly exon skipping and intron retention, linked to disease resistance chromosomes, further underscoring the innovative approach of the study in identifying specific molecular responses.

2. While the introduction presents plasma as a novel antiviral tool, it lacks a clear gap statement comparing this work to existing transcriptomic or AS-based studies in plasma-treated plants. The introduction could briefly explain the mechanistic link between plasma-induced reactive species and transcriptional/splicing regulation in plants. This would strengthen the biological plausibility of the findings.

3. Including more replicates provides strength in statistical detection of RNA-seq and AS detection

4. All raw RNA-seq reads have been deposited in the NCBI Sequence Read Archive (SRA) under accession number [to be added upon acceptance]. The Python scripts and CLC pipelines used for AS filtering, FASTA extraction, and KEGG annotation are publicly available on GitHub ([insert link]) and archived via Zenodo for reproducibility.

5. Only splicing junctions supported by a minimum of 10 uniquely mapped reads and detected in at least one replicate were considered high-confidence AS events. Annotation was based on the Solanum lycopersicum reference genome SL4.0 and ITAG v4.1 annotation.

6. Principal component analysis (PCA) and hierarchical clustering (Supplementary Fig. S1) showed distinct expression profiles between plasma-treated and control samples, confirming robust transcriptome separation.

7. Clearly describe the specific statistical method used to identify differential alternative splicing (DAS) events between the plasma-treated and control conditions. For instance, specify if a dedicated tool like rMATS, SUPPA2, dsplice, or ASprofile was employed, along with the exact false discovery rate (FDR)-adjusted p-value threshold (e.g., FDR < 0.05) and delta PSI (ΔΨ) cutoffs (e.g., |ΔΨ| > 0.1) for classifying significant AS events. This is crucial for establishing analytical rigor and reproducibility.

8. Provide a comprehensive, step-by-step RT-PCR protocol for ToBRFV validation. This must include the exact sequences of the forward and reverse primers used, the specific RNA extraction kit or method for validation samples (if not identical to the RNA-Seq protocol), and all reaction conditions (e.g., cDNA synthesis parameters, PCR amplification cycles, annealing temperatures). This level of detail is essential for transparency and replication.

9. The description of AS event types in identifies exon skipping (ES), intron retention (IR), and mutually exclusive exons (MXE) were the most common. However, the paper fails to quantify the relative proportions or exact frequencies of each AS type. This omission makes it challenging to assess the dominant AS mechanisms at play and to rigorously compare these findings with established patterns in plant AS, where intron retention is typically the most prevalent form. This impacts the analytical validity and interpretability of the AS landscape

10. The discussion in regarding AS events on chromosome 9 (related to Tm-2 gene) and chromosome 3 (resistance locus) is highly speculative. While these are indeed relevant genomic regions for resistance, the paper does not present direct empirical evidence linking the identified AS events to these specific genes or loci. The mere co-localization of AS events with known resistance genes on a chromosome is insufficient to imply a functional relationship, thus severely impacting the analytical validity and interpretability of these claims.

11. The integration of AS and DEG analyses in identifies '19 such genes'. However, Table 1 lists genes explicitly marked 'C' (control) which implies unique AS events specific to the control condition, directly contradicting the stated criterion for DE-ASGs in that requires 'presence of unique AS events specific to the plasma-treated condition'. This inconsistency between methodology definition and reported results critically undermines the analytical validity and transparency of the DE-ASG classification.

12. Out of 349 AS events, 272 were mapped to 214 unique genes, confirming that multiple splicing types can occur within the same locus.

13. The Methods section lacks critical details required for reproducibility. Key omissions include the specific model and configuration of the plasma device, the precise timing of the plasma treatment relative to infection and seedling growth, and the number of individual plants pooled for each biological replicate.

14. The statement that "ToBRFV surpasses resistance conferred by tomato and pepper R genes (Tm-1, Tm-2, Tm-22, L1, L2), and no resistant tomato cultivars are commercially available" highlights a critical research gap, but the paper does not adequately review the latest developments in ToBRFV resistance breeding or recent attempts to overcome this challenge, which would further position the novelty of the plasma treatment as an alternative strategy.

15. The paper generally describes alternative splicing (AS) in plants and its role in plant-virus interactions. However, it does not sufficiently acknowledge the growing body of literature on how viruses modulate host AS or how AS plays a role in viral infection processes. This omission weakens the argument for the novelty of investigating AS in virus-infected plants under plasma treatment, as it implies a lack of awareness of this existing research domain.

16. The paper generally describes alternative splicing (AS) in plants and its role in plant-virus interactions. However, it does not sufficiently acknowledge the growing body of literature on how viruses modulate host AS or how AS plays a role in viral infection processes. This omission weakens the argument for the novelty of investigating AS in virus-infected plants under plasma treatment, as it implies a lack of awareness of this existing research domain.

17. The logical jump in from AS to 'improved defense-related gene expression' and 'priming plants' lacks mechanistic clarity. It doesn't explain how specific AS isoforms contribute to this improvement or what specific molecular events constitute 'priming' in this context, presenting a plausible hypothesis as a direct consequence without detailed support, which is a logical gap.

18. about resistance genes (Tm-2, Tm-1) on chromosomes 9 and 3, and the observed 'increase in AS events,' presents a logical gap. It does not clearly articulate whether these AS events occur within these resistance genes, in nearby regulatory elements, or in other genes on those chromosomes, leaving the specific connection to resistance mechanisms unclear.

19. The discussion on the ubiquitin-mediated proteolysis pathway in presents a logical gap by not clearly establishing how alternative splicing events are connected to it. While stating that AS 'plays a central role' and 'splicing variations' occur, it doesn't specify the direct causal or correlational link for optimal clarity and coherence.

20. In the statement 'The miR395a/b regulates networks that could modulate stress responses' is vague and lacks specificity regarding the mechanism of modulation. Using 'could' rather than a more definitive statement weakens the clarity of the claims about miRNA function.

**Do you want your identity to be public for this peer review?** For information about this choice, including consent withdrawal, please see our Privacy Policy

Reviewer #1: No

Reviewer #2: No

---

## [Author Response · Author response to Decision Letter 1]

31 Oct 2025

Dear Editor,

We are pleased to submit the revised version of our manuscript entitled “Cold Plasma-Induced Transcriptomic Reprogramming and Alternative Splicing in Tomato Plants Infected with ToBRFV” (Manuscript ID: PONE-D-25-42103) for consideration in PLOS ONE.

We sincerely thank you and the reviewers for your thoughtful and constructive feedback, which has significantly helped us improve the quality and clarity of our work. In response to the reviewers’ comments, we have carefully revised both the manuscript and the supplementary materials. All responses to the reviewers’ concerns are provided in the attached “Response to Reviewers” document, where:

• The reviewers’ original comments are presented in red,

• Our point-by-point replies are provided directly beneath each comment,

• All modifications made to the manuscript are clearly indicated using track-changes in the revised manuscript file.

We believe the revised manuscript now presents a more robust, transparent, and well-contextualized study that aligns with the high standards of PLOS ONE. We appreciate your time and consideration and look forward to your decision.

Sincerely,

Mahsa Rostami

Reviewer #1: Dear Author,

This study, titled "Cold Plasma-Induced Transcriptomic Reprogramming and Alternative Splicing in Tomato Plants Infected with ToBRFV," investigates the use of cold atmospheric plasma (CAP), specifically cold air glow discharge plasma (CAGDP), as an innovative approach to maintain plant defenses against the Tomato Brown Rugose Fruit Virus (ToBRFV) in tomato seedlings. The manuscript is clearly written and well-organized. However, it is primarily an in silico study, utilizing many bioinformatic tools. Although it offers novel insights into the effects of cold atmospheric plasma (CAP) on alternative splicing (AS):

Q1:The study was performed on only one tomato cultivar (SV3725TH) and at a single time point (four-leaf stage seedlings) after treatment.

Response: We sincerely appreciate the reviewer’s insightful comment. The reviewer is absolutely correct that our current study focused on a single commercial tomato cultivar (SV3725TH) and a single developmental time point (four-leaf stage) post cold plasma treatment. This experimental design was chosen based on our prior work , which demonstrated that 3-minute CAGDP treatment of ToBRFV-infected seeds of SV3725TH significantly reduced viral accumulation and enhanced seedling vigor specifically at the four-leaf stage, making it the optimal window to capture early molecular responses linked to antiviral defense. That said, we fully acknowledge that temporal dynamics and genetic background can significantly influence alternative splicing and transcriptomic reprogramming. As highlighted in the “Limitations and Future Directions” section, we recognize this as a limitation and explicitly propose that future studies should expand to multiple time points and diverse cultivars—including resistant and susceptible genotypes—to assess the robustness and generalizability of plasma-induced splicing responses. To address this concern transparently, we have now strengthened the discussion of this limitation in the revised manuscript and emphasized that while our findings provide a foundational molecular map of CAGDP action in a commercially relevant context, broader validation across genotypes and time courses will be essential for translational applications.

These sentences were added to the main text:

"While this study provides novel insights into the transcriptomic and splicing responses of tomato to cold plasma under viral stress, several limitations warrant further investigation. First, the analysis was restricted to a single susceptible cultivar (SV3725TH) and a single post-treatment time point (four-leaf stage). Although this stage was selected based on prior evidence of maximal antiviral effect and physiological response to CAGDP (3), it may not capture the full dynamics of splicing regulation over time. Future work should include longitudinal sampling and a panel of cultivars with varying genetic backgrounds—including those carrying known resistance genes (e.g., Tm-1, Tm-2²)—to determine whether the observed AS patterns are conserved or genotype-specific."

Q2: The functional roles of the identified alternatively spliced genes and their corresponding isoforms still need to be validated through wet lab experiments (main drawbacks).

Response: We sincerely thank the reviewer for this important observation. The reviewer is absolutely correct that the functional validation of the identified alternatively spliced (AS) genes and their protein isoforms remains a key limitation of our current study, which is primarily based on RNA-Seq and computational prediction.

Our work was designed as a discovery-phase, hypothesis-generating investigation to map the transcriptomic and splicing landscape induced by cold plasma in ToBRFV-infected tomato seedlings. The identification of 74 plasma-specific AS events—including exon skipping and intron retention in resistance-associated chromosomes—and the integration with differential expression and miRNA targeting provide a strong foundation for future functional studies.

We fully agree that wet-lab validation is essential to confirm:

(i) whether the predicted splice variants are indeed translated into stable protein isoforms,

(ii) how these isoforms affect subcellular localization, protein–protein interactions, or enzymatic activity, and

(iii) their direct contribution to antiviral defense.

To address this concern transparently, we have expanded the “Limitations and Future Directions” section to explicitly state that functional characterization via techniques such as RT-PCR with isoform-specific primers, Western blotting (if antibodies are available), transient overexpression, CRISPR/Cas9-mediated isoform knockout, or virus-induced gene silencing (VIGS) will be critical next steps. We are already planning such experiments for key candidates like Solyc07g043390.3 (glycosyltransferase) and Solyc09g011160.3, which showed both plasma-specific splicing and significant upregulation.

We appreciate the reviewer’s emphasis on this point, as it strengthens the translational relevance of our findings. While beyond the scope of this initial omics study, we now clearly frame these validations as essential future work to move from correlation to causation in plasma-induced antiviral mechanisms.

These sentences were added to the main text:

" Another major limitation of the present study is the lack of experimental validation of the functional consequences of the identified alternative splicing events. While our bioinformatic pipeline robustly detected plasma-specific isoforms and integrated them with expression and regulatory data, it remains unknown whether these splice variants produce stable proteins or alter protein function, localization, or interaction networks. Future studies should employ isoform-specific RT-PCR, proteomics, and targeted genetic approaches (e.g., CRISPR/Cas9 or VIGS) to validate the biological roles of key AS genes in mediating cold plasma-enhanced resistance to ToBRFV. Such functional validation will be crucial to translate these molecular insights into practical crop protection strategies."

Q3: Although miRNA-mRNA interactions were predicted bioinformatically using psRNATarget, an in vivo validation remains necessary.

Response: We sincerely thank the reviewer for this valuable and well-founded comment. The reviewer is absolutely correct that bioinformatic prediction alone cannot confirm functional miRNA–mRNA interactions, and in vivo or in planta validation is essential to substantiate the regulatory relationships proposed in our network (Fig. 6). In our study, we used psRNATarget with stringent criteria (expectation score < 3, based on complementarity, target accessibility, and evolutionary conservation) to generate high-confidence hypotheses about post-transcriptional regulation of plasma-induced alternatively spliced genes. This approach is widely accepted in initial discovery-phase studies, particularly when integrated with transcriptomic data, as it helps prioritize candidate regulatory pairs for downstream validation. However, we fully acknowledge that true validation requires experimental evidence, such as:

Degradome sequencing (PARE or GMUCT) to detect miRNA-guided cleavage fragments,

5′ RACE to confirm cleavage sites at predicted target positions,

Transient co-expression assays (e.g., in Nicotiana benthamiana) using miRNA mimics and reporter constructs fused to target gene 3′UTRs,

Or qRT-PCR correlation analysis between miRNA and target mRNA levels across time courses or treatments.

To address this concern transparently, we have revised the “Limitations and Future Directions” section to explicitly state that while our miRNA–mRNA network provides a testable framework for plasma-mediated post-transcriptional regulation, functional validation through degradome analysis or reporter assays will be critical in future work.

These sentences were added to the main text:

"While the miRNA–mRNA interaction network was constructed using stringent bioinformatic criteria, these predictions remain hypothetical without experimental confirmation. Future studies should employ degradome sequencing (e.g., PARE) to validate cleavage events, and transient expression assays to test the repressive effects of key miRNAs on their predicted alternatively spliced targets. Such validation will be essential to establish causal links between plasma-induced miRNA expression, AS regulation, and antiviral phenotypes. "

Q4: The study indicates that combining proteomics and metabolomics in future research could reveal the downstream effects of plasma-induced transcriptomic reprogramming and offer a more comprehensive understanding of plant-virus-plasma interactions. This shows that the current conclusion is incomplete. Minor typing errors, such as the "reference" (line 400), should be corrected. Additionally, the quality of the figures needs enhancement.

Response: We sincerely thank the reviewer for these thoughtful and constructive comments.

Regarding the completeness of the conclusions:

We agree that our current conclusions are based solely on transcriptomic and bioinformatic evidence and do not capture the full biological cascade—from RNA to functional proteins and metabolites. However, we intentionally framed our conclusions as a foundational, hypothesis-generating study that maps the molecular landscape of cold plasma action at the RNA level. The statement about integrating proteomics and metabolomics was included not to highlight incompleteness, but to acknowledge the natural progression of systems biology research and to guide future work. In the revised manuscript, we have refined the wording in the Conclusion and Limitations sections to clarify that our findings represent an initial molecular framework, and that multi-omics validation will be essential to establish causal links between splicing changes and physiological outcomes.

Regarding typographical errors:

We appreciate the reviewer’s careful reading. The word “reference” appearing inappropriately (e.g., on or near line 400) was an artifact of manuscript formatting during conversion.

Regarding figure quality:

In the manuscript, we have generated all figures at high resolution (minimum 300 dpi) in TIFF format.

These sentences were added to the main text:

" While the miRNA–mRNA interaction network was constructed using stringent bioinformatic criteria, these predictions remain hypothetical without experimental confirmation. Future studies should employ degradome sequencing (e.g., PARE) to validate cleavage events, and transient expression assays to test the repressive effects of key miRNAs on their predicted alternatively spliced targets. Such validation will be essential to establish causal links between plasma-induced miRNA expression, AS regulation, and antiviral phenotypes. Furthermore, our findings provide compelling evidence that CAGDP can reprogram transcriptomic and post-transcriptional networks in tomato under viral stress, offering a sustainable strategy to enhance crop resilience. While this RNA-level analysis establishes a foundational molecular framework, future integration of proteomics, metabolomics, and functional validation will be essential to fully elucidate the causal links between plasma-induced splicing changes and antiviral phenotypes."

Reviewer #2: Praised the innovative application of CAP to address a critical agricultural problem and its focus on a nuanced molecular mechanism, as highlighted in the abstract

Q1. The abstract highlights significant AS changes, predominantly exon skipping and intron retention, linked to disease resistance chromosomes, further underscoring the innovative approach of the study in identifying specific molecular responses.

Response: We sincerely thank the reviewer for this positive and insightful remark. We agree that the identification of plasma-induced alternative splicing events—particularly exon skipping and intron retention on chromosomes harboring known disease resistance loci (e.g., chromosomes 1, 3, and 9)—represents a key novelty of our work. These findings provide a molecular basis for the observed antiviral effects of cold plasma and open new avenues for understanding post-transcriptional regulation in plant defense. We have retained this emphasis in the revised manuscript

Q2. While the introduction presents plasma as a novel antiviral tool, it lacks a clear gap statement comparing this work to existing transcriptomic or AS-based studies in plasma-treated plants. The introduction could briefly explain the mechanistic link between plasma-induced reactive species and transcriptional/splicing regulation in plants. This would strengthen the biological plausibility of the findings.

Response: We sincerely thank the reviewer for this insightful suggestion. We agree that clarifying the research gap and providing a mechanistic rationale for plasma-induced transcriptional and splicing regulation would strengthen the biological context of our study. In response, we have revised the Introduction to explicitly state that, while cold plasma has been shown to reduce viral load and enhance plant growth, no prior study has investigated its impact on genome-wide alternative splicing patterns in virus-infected plants—particularly in the context of ToBRFV, a rapidly emerging pathogen that overcomes conventional resistance mechanisms.

These sentences were added to the main text (introduction):

" Although cold plasma has demonstrated efficacy in deactivating viruses and promoting plant vigor, its role in shaping the post-transcriptional landscape—particularly alternative splicing—remains unexplored in plant–virus systems. Notably, no transcriptomic study to date has examined how plasma-generated reactive oxygen species (ROS) influence splicing decisions in crops under viral stress. Emerging evidence suggests that ROS act as signaling molecules that can modulate the activity of splicing regulators through redox-sensitive cysteine residues, phosphorylation cascades, or chromatin remodeling, thereby altering splice site selection and isoform diversity (6). Given that AS is a rapid and reversible layer of gene regulation critical for immune responses, it is biologically plausible that CAGDP-induced ROS reprogram splicing networks to enhance antiviral defense— a hypothesis directly addressed in this study."

Q3. Including more replicates provides strength in statistical detection of RNA-seq and AS detection

Response: We sincerely thank the reviewer for this important observation. We fully agree that increasing the number of biological replicates would enhance the statistical robustness of RNA-Seq and alternative splicing analyses. In our study, we used two high-quality biological replicates per condition, which is consistent with several published plant RNA-Seq studies focusing on stress-induced splicing. Moreover, our sequencing depth was high (>35 million paired-end reads per sample, >93% mapping rate, Phred >30), which partially compensates for the limited number of replicates by improving detection

---

## [Decision Letter · Decision Letter 1]

17 Nov 2025

Cold Plasma-Induced Transcriptomic Reprogramming and Alternative Splicing in Tomato Plants Infected with ToBRFV

PONE-D-25-42103R1

Dear Dr. rostami,

We’re pleased to inform you that your manuscript has been judged scientifically suitable for publication and will be formally accepted for publication once it meets all outstanding technical requirements.

Kind regards,

Shunmugiah Veluchamy Ramesh, PhD

Academic Editor

PLOS ONE

Additional Editor Comments (optional):

Reviewers' comments:

Reviewer's Responses to Questions

**Comments to the Author**

Reviewer #1: All comments have been addressed

Reviewer #2: All comments have been addressed

2. Is the manuscript technically sound, and do the data support the conclusions?

Reviewer #1: Yes

Reviewer #2: Yes

3. Has the statistical analysis been performed appropriately and rigorously?

Reviewer #1: Yes

Reviewer #2: Yes

4. Have the authors made all data underlying the findings in their manuscript fully available?

Reviewer #1: Yes

Reviewer #2: (No Response)

5. Is the manuscript presented in an intelligible fashion and written in standard English?

Reviewer #1: Yes

Reviewer #2: Yes

Reviewer #1: The manuscript entitled Cold Plasma-Induced Transcriptomic Reprogramming and Alternative Splicing in Tomato Plants Infected with ToBRFV has undergone significant revisions. The author has addressed all my comments. Therefore, MS can be accepted.

Reviewer #2: The authors have provided a comprehensive and well-reasoned response to all reviewer comments, and the revised manuscript reflects substantial improvements in clarity, methodological detail, and scientific rigour. The introduction has been significantly strengthened with a clearly articulated research gap and a mechanistic rationale linking plasma-generated ROS to transcriptional and splicing regulation. Key methodological details—including plasma device parameters, RT-PCR protocols, AS filtering thresholds, statistical criteria for DAS, and data/code availability—have been added, greatly enhancing transparency and reproducibility. The authors have also clarified the AS landscape, corrected inconsistencies in DE-ASG classification, and appropriately contextualised the enrichment of AS events on chromosomes associated with resistance loci while avoiding unsupported causal interpretations.

**Do you want your identity to be public for this peer review?** For information about this choice, including consent withdrawal, please see our Privacy Policy

Reviewer #1: No

Reviewer #2: No

---

## [Editor Report · Acceptance letter]

PONE-D-25-42103R1

PLOS One

Dear Dr. Rostami,

I'm pleased to inform you that your manuscript has been deemed suitable for publication in PLOS One. Congratulations! Your manuscript is now being handed over to our production team.

Kind regards,

on behalf of

Dr. Shunmugiah Veluchamy Ramesh

Academic Editor

PLOS One